# The Interactive Effect of Genetic and Epigenetic Variations in *FKBP5* and *ApoE* Genes on Anxiety and Brain EEG Parameters

**DOI:** 10.3390/genes13020164

**Published:** 2022-01-18

**Authors:** Irina L. Kuznetsova, Natalya V. Ponomareva, Ekaterina A. Alemastseva, Andrey D. Manakhov, Tatyana V. Andreeva, Fedor E. Gusev, Evgeny I. Rogaev

**Affiliations:** 1Center for Genetics and Life Science, Sirius University of Science and Technology, 354349 Sochi, Russia; irakuzn@gmail.com (I.L.K.); manakhov@rogaevlab.ru (A.D.M.); andreeva@rogaevlab.ru (T.V.A.); gusevfe@gmail.com (F.E.G.); 2Laboratory of Evolutionary Genomics, Department of Genomics and Human Genetics, Vavilov Institute of General Genetics, Russian Academy of Sciences, 119333 Moscow, Russia; alemasceva@mail.ru; 3Research Center for Neurology, 125367 Moscow, Russia; ponomareva@neurology.ru; 4Centre for Genetics and Genetic Technologies, Faculty of Biology, Lomonosov Moscow State University, 119192 Moscow, Russia; 5Department of Psychiatry, UMass Chan Medical School, 222 Maple Ave, Reed-Rose-Gordon Building, Shrewsbury, MA 01545, USA

**Keywords:** *FKBP5*, ApoE, anxiety, electroencephalography (EEG), DNA methylation

## Abstract

*FKBP51* is a key stress-responsive regulator of the hypothalamic–pituitary–adrenal axis. To elucidate the contribution of rs1360780 *FKBP5* C/T alleles to aging and longevity, we genotyped *FKBP5* in a cohort of 800 non-demented and Alzheimer’s disease (AD) subjects of different age, taking into account the allele state of *ApoE* ε4, the major risk factor for AD. Furthermore, we searched for the association of *FKBP5* with subcohorts of non-demented subjects evaluated for anxiety and resting-state quantitative EEG characteristics, associated with cognitive, emotional, and functional brain activities. We observed that increased state anxiety scores depend on the combination of the *FKBP5* and *ApoE* genotypes and on the DNA methylation state of the *FKBP5* promoter and *ApoE* genotype. We also found a significant gender-dependent correlation between *FKBP5* promoter methylation and alpha-, delta-, and theta-rhythms. Analysis of the *FKBP5* expression in an independent cohort revealed a significant upregulation of *FKBP5* in females versus males. Our data suggest a synergistic effect of the stress-associated (*FKBP5*) and neurodegeneration-associated (*ApoE*) gene alleles on anxiety and the gender-dependent effect of *FKBP5* on neurophysiological brain activity.

## 1. Introduction

The *FKBP5* gene encodes the FKBP51 protein, which is a co-chaperone of the hsp90 protein involved in the regulation of the hypothalamic–pituitary–adrenal axis as a component of the oligomeric complex of the glucocorticoid receptor. The presence of the T variant in SNP rs1360780 (C > T) in the second intron of the *FKBP5* gene has been associated with a significant increase in the risk of developing neuropsychiatric disorders [1,2,3,4]. C/T variation could affect the three-dimensional DNA structure by forming a TATA box-like sequence, which in turn can contribute to the transcription of the corresponding mRNA and finally lead to epigenetic modifications of *FKBP5*. This hypothesis was first supported by the genotype–methylation analysis of *FKBP5* in the blood and saliva of adult victims of childhood trauma [5] and was reinforced by the analysis of other cases of child abuse [6,7]. The methylation level within the *FKBP5* gene from the buccal epithelia of children and blood samples of adults responding to the psychological treatment changed during the therapy for T-allele carriers, leading to higher treatment efficacy [8,9]. Intriguingly, several studies of patients exposed to childhood maltreatment did not confirm an association between genetic-dependent epigenetic changes in *FKBP5* and psychiatric deviations [10,11]. Genotype-independent alterations in methylation of a particular CpG within *FKBP5* in blood were found in Holocaust victims and their offspring [12]. Thus, it is reasonable to expect that some additional factors contributing to the complex interplay between genotype, DNA methylation, and psycho-emotional status may exist. 

There are substantial reasons to believe that *FKBP5* influences brain activity and that changes in *FKBP5* expression may contribute to brain function. Attempts to relate brain activity and structure have revealed that T-allele carriers are more sensitive to psychological trauma, leading to abnormalities in the volume or function of some brain regions [13,14,15,16,17]. During aging, humans become less stress-resistant. Analysis of the methylation status of CpG nucleotides around the *FKBP5* locus using a 450 K methylation panel (Illumina) revealed two CpGs showing age-dependent methylation in blood [18]. The same tendency was previously demonstrated for CpG methylation within the fifth intron of *Fkbp**5* in mice [19]. Recently, our group used ChIP-seq analysis of brain samples from neurons of the prefrontal cortex and found age-dependent epigenetic variations within the human *FKBP5* gene [20]. Notably, age-related epigenetic changes were mostly confined to the region downstream of the *FKBP5* promoter, which is absent from the 450 K methylation array, and explains the scarcity of publicly available methylation data for this region.

Dynamic epigenetic changes across *FKBP5* in the brain and association of the risk allele with brain pathology point to the possible role of *FKBP5* in pathogenesis of age-dependent neurodegeneration such as Alzheimer’s disease (AD). Neuropsychiatric disorders, such as depression and posttraumatic stress disorder (PTSD), are also known to promote AD progression [21,22]. The major known risk factor of AD, the *ApoE* ε4 allele, is not associated with depression in non-demented individuals [23], while the study of AD patients revealed the over-representation of *ApoE* ε4 among AD women with depression [24]. Furthermore, PTSD has been shown to play a critical role in driving more severe cognitive dysfunction in both *FKBP5* rs1360780 T-allele and *ApoE* ε4 carriers [25]. Studies aimed at uncovering gene variants involved in neuropsychiatric disorders in AD patients revealed the associations of *MAOA, BDNF, TPH1*, and *FKBP5* genes with AD-associated depression [26].

We attempted for the first time to combine in one study the analysis of the effects of rs1360780 polymorphism in the *FKBP5* gene and ε2/ε3/ε4 polymorphisms in the *ApoE* gene on anxiety levels and EEG parameters. We evaluated the influence of these genotypes on the resting-state quantitative electroencephalogram. EEG rhythms reflect the neural processes underlying human cognition. The results of EEG studies imply the relevance of quantitative EEG characteristics, in particular, the relative power of EEG frequency bands (delta (<4 Hz), theta (~4–8 Hz), alpha (8–13 Hz), beta (12–30 Hz)) as potential biomarkers of AD and psychotic disorders [27,28,29,30]. Finally, we analysed the association of the *FKBP5* promoter methylation status with the anxiety level and EEG features in carriers of both risk alleles.

## 2. Results

### 2.1. Aging and rs1360780 Alleles

Allele frequencies of *FKBP5* rs1360780 in the ND, AD, and Nona+ groups are summarised in Table 1. There were no significant differences in rs1360780 allele frequencies between the groups. Similarly, no correlation between the age of non-demented individuals (including Nona+) and T/C allele frequencies was observed. To analyse the possible effect of rs1360780 on the development of depression, a group of AD patients was subdivided into two subgroups: AD subjects with depression and AD subjects without depression. Our data are consistent with a weak tendency of increased frequency of the T allele in rs1360780 in the group of patients with AD and depression. However, this tendency was not statistically significant (*p* = 0.25311).

We also genotyped the polymorphisms ε2/ε3/ε4 in the *ApoE* gene in all groups (Appendix A). As expected, significant differences were revealed in the allele frequencies of polymorphisms in *ApoE* in the AD group compared to the ND and Nona+ groups. The frequency of the ε4 allele was significantly higher in patients with AD, while the frequency of the ε2 allele was significantly lower in AD patients than in the ND and Nona+ patients.

### 2.2. FKBP5 rs1360780 and ApoE ε2/ε3/ε4 in Association with Anxiety

Our analysis of the effect of rs1360780 C/T polymorphism in *FKBP5* on the degree of anxiety in the non-demented subgroup characterised by the Spielberger test showed that there were no differences between the average quantitative indicators of the degree of both trait and state anxiety in all selected groups (for trait anxiety, Figure 1a, *p* = 0.307809; for state anxiety, Figure 1b, *p* = 0.885592). Considering the possible contribution of sex and T allele in rs1360780 to the anxiety profiles, we did not observe significant differences in state and trait anxiety scores (H (3, *n* = 167) = 3.654349, *p* = 0.3013 and F(1, 164) = 0.59507, *p* = 0.441576, respectively). Likewise, for the ε4 allele in the *ApoE* gene, there were no statistically significant differences (for trait anxiety, Figure 1c, *p* = 0.508077; for state anxiety, Figure 1d, *p* = 0.111557). When the data were analysed separately for women and men, no contribution of the ε4 allele on the state and trait anxiety scores was found (F(1, 163) = 0.11828, *p* = 0.731355 and F(1, 164) = 0.01809, *p* = 0.893185).

Next, we analysed the effect of the simultaneous presence of the T and ε4 alleles in rs1360780 in the *FKBP5* and *ApoE* genes, respectively, on the level of state and trait anxiety in individuals from the ND group. We found a statistically significant increase in state, but not trait anxiety scores in T-(CT&TT genotypes) and ε4 allele carriers compared to the carriers of the T allele, but not the ε4 allele in ND groups (F(1, 163) = 4.5187, *p* = 0.0350). It remained significant after Bonferroni correction (**p*-value = 0.049432; Figure 1f). The analysis of the cumulative effect of T and ε4 alleles on the state and trait anxiety scores in groups of women and men was not performed because of the small size of the patient groups.

### 2.3. rs1360780 and ε4 Polymorphisms in Association with the Electrical Activity of the Brain

Case–control studies of brain electrical activity in people with various neuropsychiatric disorders, as well as people with AD, demonstrated that the course of these diseases is accompanied by distortions in the electrical activity of the brain [31,32].

Our statistical examination of the normalised EEG relative power values revealed a significant interaction between the *FKBP5* genotype, EEG bands, and sex. Post hoc comparison showed that the relative power of delta and beta2 bands was higher in men with *FKBP5* CT&TT, while the alpha relative power in men with CT&TT genotypes tended to be lower than that in men with the CC genotype (Figure 2). There was no significant difference between women with the *FKBP5* CC and *FKBP5* CT&TT genotypes.

### 2.4. Interplay between FKBP5 Promotor Methylation and Anxiety

The methylation levels of the three CpGs in the *FKBP5* promoter region were not associated with state or trait anxiety in 42 tested individuals. There was also no age- or sex-dependent methylation of these CpGs. When the *FKBP5* genotype was considered as a possible contributing factor, we observed a statistically significant correlation between higher methylation at CpG1 and trait anxiety (r = 0.44, *p* = 0.00893). The same correlation was observed for CpG1 and *ApoE* ε4− (r = 0.34, *p* = 0.01849). When analysed together (CC+ and *ApoE* ε4−), the correlation was even more pronounced (r = 0.58, *p* = 0.00220) (Figure 3).

### 2.5. FKBP5 Methylation and EEG

#### 2.5.1. Delta Rhythm

In the carriers of the *FKBP5* CC genotype, methylation levels within the CpG2 promoter region of the *FKBP5* gene correlated negatively with the log-transformed relative power of delta activity (r = −0.37, *p* = 0.04, *n* = 32). In the male and female subgroups, when considered separately, the correlation was below the significance level (*p* < 0.05). In the carriers of the *FKBP5* CT&TT genotypes, this correlation was not significant.

#### 2.5.2. Theta Rhythm

In the entire sample of individuals with different *FKBP5* genotypes, the methylation level within CpG1 residing in the promoter region of the *FKBP5* gene correlated negatively with the log-transformed relative power of the theta activity (r = −0.28, *p* = 0.03, *n* = 59). However, in the groups with *FKBP5* CC and *FKBP5* CT&TT genotypes, when considered separately, the correlation did not reach the significance threshold (*p* < 0.05).

In carriers of the *ApoE* ε4− genotype, but not in the carriers of the *ApoE* ε4+ genotype, the negative correlation between the methylation level within the CpG1 promoter region of the *FKBP5* gene and the log-transformed relative power of the theta activity was significant (r = −0.41, *p* = 0.007, *n* = 41).

#### 2.5.3. Alpha Rhythm

In the carriers of the *FKBP5* CC genotype, the methylation level within CpG2 correlated positively with the log-transformed relative power of the Alpha activity (r = 0.37, *p* = 0.04, *n* = 32). When considered separately for men and women, this correlation was observed in women (r = 0.42, *p* = 0.04, *n* = 24) but not in the male subgroup (Figure 4).

#### 2.5.4. Beta1 and Beta2 Rhythm

Significant correlation with *FKBP5* methylation was not observed.

### 2.6. Meta-Analysis of FKBP5 Expression in the Blood and Brain

To understand whether sex and age could predict the levels of *FKBP5* expression, the data for 755 whole blood and 255 brain cortex samples from the GTEx project were extracted and subjected to a multiple linear regression analysis. In both blood and cortex, the model was found to be statistically significant (F(2, 752) = 29.37, *p*-value = 5.215 × 10^−13^ for blood and F(2, 252) = 6.618, *p* = 0.00158 for cortex). In both models, age was a significant predictor (*p*-value = 3.71 × 10^−12^ for blood and *p*-value = 0.004 for cortex) with opposing effects: *FKBP5* expression decreased with age in blood but increased in the cortex. Sex was also a significant predictor in both blood (*p* = 0.004) and brain (*p* = 0.02); in both the blood and brain, female subjects tended to have higher *FKBP5* expression than male subjects (Figure 5).

## 3. Discussion

In this study, we analysed the frequencies of the rs1360780 polymorphism in the *FKBP5* gene and ε2/ε3/ε4 polymorphisms in the *ApoE* gene in non-demented people from different age population samples, and nona+centenarians, as well as people with AD. We did not find differences in the frequency of alleles or genotypes of the rs1360780 polymorphism in the *FKBP5* gene when comparing all the groups in the study. Based on the results obtained, it could be concluded that this polymorphism does not contribute to longevity if we consider it regardless of the action of the environment or interaction with other genetic factors.

Since the T allele of rs1360780 in the *FKBP5* gene was associated not only with depressive behavior but also with the development of various mental diseases, we decided to study the effect of this polymorphism on the level of anxiety. Previous studies have reported that the level of anxiety of the individual can be determined genetically to a certain extent [33,34]. The results of our study showed that the level of state anxiety on the Spielberger scale was higher in carriers of the T and ε4 alleles from a population of different ages in the *FKBP5* and *ApoE* genes, respectively, than in carriers of the T allele who lack the ε4 allele. Previously, it was established that the concentration of the main hormone response to stress, cortisol, in the cerebrospinal fluid was higher in AD patients homozygous for the ε4 allele in the *ApoE* gene than in the carriers of other genotypes suffering from AD [35]. Moreover, the *ApoE* ε4 and SNP rs1360780 T alleles were associated with the severity of PTSD [36,37]. Combined with these data, our results suggest a potential synergistic effect of the studied alleles on the level of state anxiety.

Our study shows that the *FKBP5* polymorphism rs1360780 is associated with the delta and beta2 relative power in the resting-state EEG of non-demented men. We showed an increase in the relative power of delta and beta2 bands in men carrying the *FKBP5* CT&TT variants. We also found a tendency for lower alpha relative power for men carrying *FKBP5* CT&TT variants compared with men with the *FKBP5* CC variant. These effects of the *FKBP5* genotype on EEG in men were independent of age.

An increase in delta relative power combined with a reduction in alpha power is a sign of an inappropriate arousal state, which leads to a reduced ability to attend to relevant information [28]. Slowing of EEG is a uniform finding in AD [27,29,30]. In mice, the chaperone FKBP51 can work with Hsp90 to produce tau protein, facilitating its neurotoxicity [38]. However, an increase in delta power in the carriers of *FKBP5* CC genotype was age-independent, combined with an increase in the beta2 power, and very mild, and it is unlikely that such alterations are caused by early AD-related pathology. A moderate increase in EEG delta power combined with an increase in beta power was previously found in patients with major depression, bipolar disorders, and in individuals with various forms of disinhibitory psychopathology [28,39,40]. Elevated beta power was found to be associated with alterations in inhibitory controls and anxiety [41,42]. These EEG alterations were suggested to represent a candidate endophenotype for such disorders, as they were also detected in the unaffected relatives of the patients [39]. Previously, we reported an increase in beta-activity in subjects with the AD-risk *PICALM* GG genotype, with the effect being more pronounced in subjects older than 50 years of age [43].

A significant overrepresentation of the *FKBP5* TT and TC genotypes compared to the CC genotype has been revealed in individuals presenting with major depression [44]. Such differences were found to be more pronounced in men than in women [45]. Our findings on the association of the *FKBP5* genotype with EEG characteristics in men are consistent with these results. As already mentioned, the Hsp90 co-chaperone *FKBP5* is a regulator of glucocorticoid receptor complex (GR) activity [46]. GR activity terminates physiological stress responses mediated by glucocorticoid hormones. The gender specificity in our data may be explained by the functional binding sites for sex steroids in *FKBP5* [47].

The results suggest that EEG alterations in men with the *FKBP5* TT/CT genotype are indicative of the endophenotype predisposing to the inappropriate arousal state, cortical disinhibition, and/or neuronal hyperexcitability, which may be related to high levels of anxiety or other kinds of disinhibitory psychopathology.

The results of this study show that the demethylation of several CpGs in the promoter region of *FKBP5* is associated with a decrease in alpha EEG relative power and an increase in delta and theta relative power in healthy adults. The correlation was moderate and was observed for delta and alpha relative power in the carriers of *FKBP5* CC, but not in the *FKBP5* CT&TT carriers. The correlation of *FKPB5* demethylation with theta relative power was found in *ApoE* ε4 carriers, but not in *ApoE* ε4+ carriers. A moderate reduction in alpha power combined with an increase in delta and theta relative power is a feature of an inappropriate arousal state [28]. Such EEG features may be related to the effect of *FKBP5* on stress vulnerability. 

Demethylation of the *FKBP5* gene resulting in upregulated expression has been implicated in the response to stress, which can lead to impaired glucocorticoid-mediated negative feedback of the HPA axis and persistently elevated cortisol levels [48,49]. Long-term epigenetic responses appear to predispose some individuals to psychiatric disorders such as depression and PTSD, where a moderate decrease in alpha relative power and an increase in slow-wave relative power have been previously detected [28]. The findings of the present study help to disentangle the complex interplay between *FKBP5* methylation and brain functional alterations.

Our results show a number of interesting sex-specific correlations, but the subgroups encompassed fewer than 100 subjects and were limited to blood cells. Many other studies have confirmed the sex-specific influence of variations within the *FKBP5* gene. Previously, our group performed H3K4me3 profiling of neurons from the prefrontal complex and uncovered the gender-dependent activity of the *FKBP5* gene [20]. Here, we analysed the GTEx project database for *FKBP5* expression in the blood and brain cortex and confirmed a significant difference.

Thus, we demonstrate for the first time the gender-specific effect of the rs1360780 polymorphism in the *FKBP5* gene on the characteristics of the electrical activity of the brain and the development of anxiety, both influenced by the allelic polymorphism in the *ApoE* gene. Authors should discuss the results and how they can be interpreted from the perspective of previous studies and of the working hypotheses. The findings and their implications should be discussed in the broadest context possible. Future research directions may also be highlighted.

## 4. Materials and Methods

### 4.1. Human Samples

Venous blood samples of non-demented adults (ND), up to 89 years old, and Nona+ (≥90 years old) were collected for this study (Table 2). To compare the frequency of rs1360780 and *ApoE* ε4 alleles in patients with dementia, we used DNA from patients with AD [50]. All individuals from the groups belonged to the Russian population, predominantly from the Moscow area. Informed consent was provided by all subjects or their caregivers.

A total of 168 individuals from the non-demented (ND) group were characterised by the level of state and trait anxiety using the Spielberger scale. A total of 161 individuals from the same group were also characterized by the level of alpha, beta1, beta2, delta, and theta rhythmic activity of the brain using the electroencephalography followed by quantitative analysis. Patients diagnosed with clinically probable AD according to the NINCDS-ADRDAAD and 188 individuals from the AD group were characterised for depression based on the NPI/NH tests.

### 4.2. Genotyping

DNA samples isolated from the venous blood of individuals were genotyped for the rs1360780 *FKBP5* and ε2/ε3/ε4 polymorphisms in the *ApoE* gene.

Polymorphism rs1360780 in the *FKBP5* gene was detected by quantitative PCR TaqMan analysis. Specifically, we used a set of primers and probes (C-FAM and T-VIC) provided by the “DNA-Synthesis” (Moscow, Russia) and Isogen set for the PCR Gen Pack Real-Time PCR “Core” (Moscow, Russia) using the 7500 Real-Time PCR System (Applied Biosystems, Waltham, MA, USA). ApoE genotyping was performed using RFLP analysis, as previously described [50].

### 4.3. Methylation

Sodium bisulfite conversion was performed with 300 ng of DNA using the EZ Methylation Kit (Zymo Research, Irvin, CA, USA). Forward and reverse primers (Appendix A) and the PyroMark PCR kit (Qiagen, Hilden, Germany) were used to amplify the *FKBP5* promoter region (chr6:35656792–35656628, UCSC NCBI37/h19). PCR products were sequenced using the PyroMark 48 system (Qiagen). The percentage DNA methylation of three target CpGs (CpG1 chr6:35656662, CpG2 chr6:35656666, CpG3 chr6:35656668) was quantified using PyroMarkCpG software.

### 4.4. EEG Procedure

The registration and evaluation of EEG was carried out in accordance with the IPEG guidelines. All recordings were obtained in the afternoon at 3–4 p.m. EEGs were recorded for 3 min during resting, the subjects sitting comfortably in a chair. They were asked to close their eyes and relax but stay awake during the recording. Prior to each recording, subjects were instructed to relax but stay awake. In order to keep the level of vigilance constant, an experimenter controlled the subject and the EEG traces online. He verbally alerted the subject any time there were signs of behavioral and/or EEG drowsiness.

EEG was recorded with a Nihon Kohden 4217 G EEG using the time constant of 0.3 s. The high-frequency cut-off was 45 Hz. The 14 Ag/AgCl electrodes were placed according to the international 10–20 system at the O2, O1, P4, P3, C4, C3, F4, F3, Fp2, Fp1, T4, T3, F8, and F7 positions. Linked ears served as the reference. Electrode impedance did not exceed 10 kΩ. During the recording, 180 s of EEG at rest was simultaneously sampled at 256 Hz per channel and stored on a computer for further analysis offline. The EEG was reviewed visually for artifacts. Periods of artifact were eliminated from subsequent analysis. After the elimination of artifacts, segments of resting EEG of 120 s duration were selected for further analysis.

### 4.5. EEG Analysis

Frequencies below 2 Hz and above 35 Hz were eliminated by digital filtering. Thirty 4 s epochs free of artifacts of resting EEG were processed by fast Fourier transform. The relative powers (% of total EEG power) of the delta (2.00–3.99 Hz), theta (4.00–7.99), alpha (8.00–12.99), beta1 (13.00–19.99), and beta2 (20.00–30.00) bands and for the regions occipital (O2, O1), parietal (P4, P3), central (C4, C3), frontal 1 (F4, F3), frontal 2 (Fp2, Fp1), mid-temporal 2 (T4, T3) and anterior temporal (F8, F7) were calculated. Log transformations of the relative power of the various bandwidths in each derivation were calculated in order to compensate for data skewedness, as recommended by [51], using log [*x*/(1 − *x*)], where *x* is the fraction of total power for each 4 s sample. The average log relative power for each frequency band was then calculated. The details of the spectral analysis procedures have been previously described [43].

### 4.6. Statistical Analysis

Differences in allele frequencies of *FKBP5/ApoE* polymorphisms in all groups were compared using Pearson’s chi-squared test. The Shapiro–Wilk test, Fisher’s test, and Levene’s test were applied to check the normality and homogeneity of variances, respectively, for all groups and subgroups. Differences in the anxiety scores and alpha relative power in the groups of ND individuals were compared using Student’s *t*-test in the case of two samples and one-way analysis of variance (ANOVA) in the case of more than two samples under the condition of normally distributed samples followed by post hoc analysis (Bonferroni correction, Duncan’s test).

If the studied samples did not follow a normal distribution, the Mann–Whitney *U* test (Mann–Whitney *U*-test) (comparison of two samples) or the Kruskal–Wallis test (Kruskal–Wallis one-way analysis of variance) when comparing more than two samples.

### 4.7. FKBP5 Expression Analyses

Multiple regression analysis of *FKBP5* expression was performed using GTEx release v8 RNA-seq data. The *FKBP5* transcript per million (TPM) value was used as the dependent variable. In the deidentified open-access GTEx, the subject metadata ages are reported to be rounded up to the decade. Therefore, the start of the decade was used as an independent variable in the model. The other variable, sex, was used without any modifications. For the analysis of blood, only samples labelled with whole blood were considered. For brain analysis, only samples labelled with “Brain-Cortex” were considered. The analysis was performed using R software (R Core Team 2020. R: Language and Environment for Statistical Computing. R Foundation for Statistical Computing, Vienna, Austria. URL https://www.R-project.org/ (accessed on 12 December 2021)).

## Figures and Tables

**Figure 1 genes-13-00164-f001:**
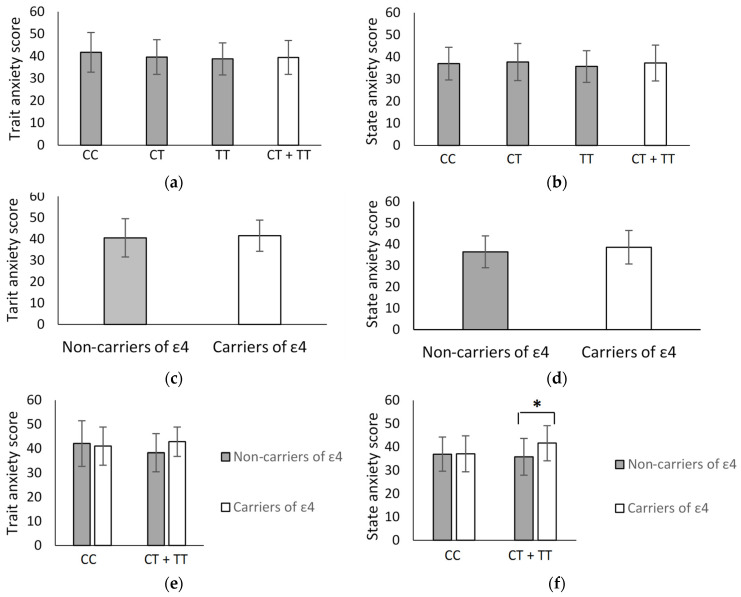
Means ± SD of trait (**a**) and state (**b**) anxiety scores in the subgroups of CC, CT, and TT genotype carriers (grey bars) and T allele carriers (white bars) in the *FKBP5* gene in subjects from the ND group. Means ± SD of trait (**c**) and state (**d**) anxiety scores in the subgroups of ε4 allele non-carriers (grey bar) and ε4 allele carriers (white bar) in the *ApoE* gene in the ND group. There is no significant difference in the state anxiety score in the given groups. Means ± SD of trait (**e**) and state (**f**) anxiety scores in T ± and ε4 ± allele carriers. The significant difference (* *p* = 0.007202) in state anxiety scores between T/ε4− and T/ε4+ allele carriers.

**Figure 2 genes-13-00164-f002:**
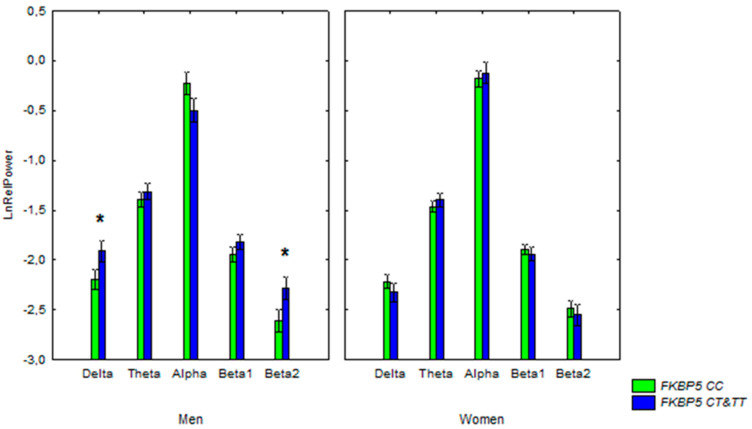
Log-transformed relative power (mean and SE) of EEG bands in the non-demented men and women with the *FKBP5* CC and *FKBP5* CT&TT genotypes, *: *p* < 0.05.

**Figure 3 genes-13-00164-f003:**
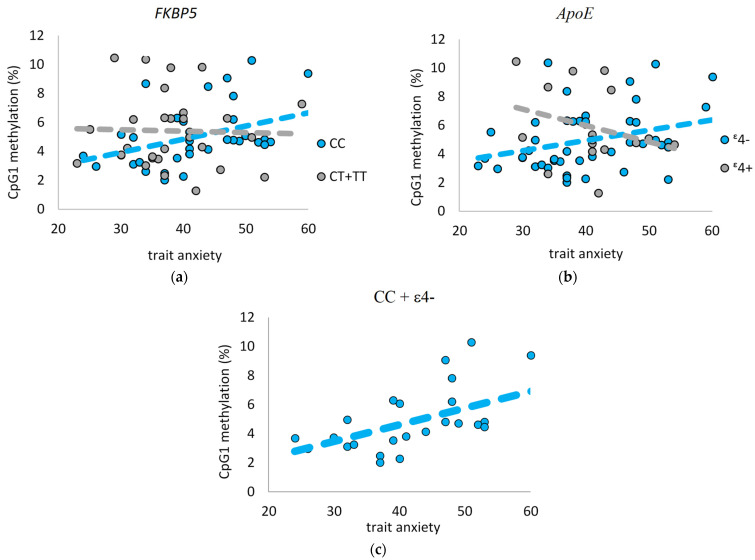
CpG1 methylation level (%) of the *FKBP5* promoter region for CC/CC + CT rs1360780 (**a**) and *ApoE* ε4−/ε4+ (**b**) carriers and elevation of trait anxiety. The additive effect of CC and ε4− carries (r = 0.58, *p* = 0.00220) (**c**).

**Figure 4 genes-13-00164-f004:**
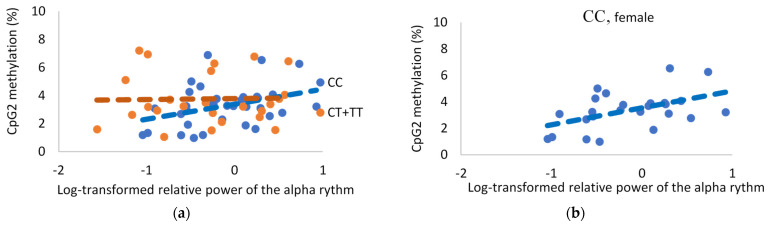
The DNA methylation status of the CpG2 found in the *FKBP5* promotor region is positively correlated with the relative power of Alpha rhythm for rs1360780 CC carriers (**a**). The correlation is higher for female CC carriers (**b**).

**Figure 5 genes-13-00164-f005:**
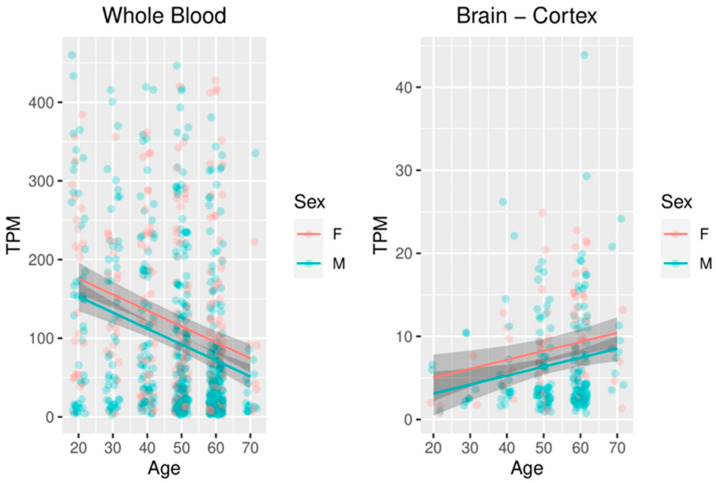
Scatter plots and linear regression lines (with 95% confidence intervals) of *FKBP5* expression vs. age in the whole blood and cortex samples of the GTEx project.

**Table 1 genes-13-00164-t001:** rs1360780 allele frequencies and genotype frequencies in study participants.

Group	Genotype Frequency			Allele Frequency	
	CC	CT	TT	C	T
ND	0.582	0.353	0.065	0.759	0.241
Nona+	0.580	0.370	0.050	0.765	0.235
Patients with AD (total)	0.556	0.367	0.077	0.740	0.260
Patients with AD without depression	0.617	0.290	0.093	0.762	0.238
Patients with AD with depression	0.495	0.434	0.071	0.712	0.288

**Table 2 genes-13-00164-t002:** Demographic characteristics of AD and control groups.

Group	Number			Age Mean ± SD (Age Range)		
	Total	Females	Males	Total	Females	Males
ND	479	214	265	62.07 ± 18.03 (19–89)	58.23 ± 16.77 (19–89)	65.17 ± 18.45 (19–88)
Nona+	100	76	24	97.00 ± 4.49 (90–107)	97.30 ± 4.48 (90–107)	97.00 ± 4.60 (90–105)
AD	221	156	65	67.00 ± 8.75 (41–87)	66.82 ± 9.12 (41–87)	68.00 ± 7.64 (52–84)

## Data Availability

Not applicable.

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
