# Peer review of "The Interactive Effect of Genetic and Epigenetic Variations in FKBP5 and ApoE Genes on Anxiety and Brain EEG Parameters"

_genes, 2022, doi:10.3390/genes13020164_

Round 1

Reviewer 1 Report

The T variant in SNP of FKBP5 is shown to associate with the risk of psychiatric disorders in humans. Here, Kuznetsova et al expands this finding by providing previously unknown association  of increased state of anxiety (based on EEG parameters) to a synergistic effect of FKBP5 C/T alleles and ApoE gene alleles. The paper is interesting and the data appears to be interpreted correctly.

Minor comment: Please correct typo in Line 121 (reference to Fig.2F should be instead Fig.1F).     

Author Response

Minor comment: Please correct typo in Line 121 (reference to Fig.2F should be instead Fig.1F). 

Response: The missprint  "Fig.2F"  instead "Fig.1F" was corrected (line 121, yellow highlight). Please see attachment.

Reviewer 2 Report

The manuscript “The interactive effect of genetic and epigenetic variations in
FKBP5 and APOE genes on anxiety and brain EEG parameters” by Kuznetsova
et al. gives an interesting perspective on the interaction of genetic and epigenetic
factors of dementia and anxiety, and their reflection of brain-electric activity.
Anxiety and EEG metrics may have diagnostic and prognostic value to dementia
in the future and this work presents some important data that may guide future
studies.
I think the manuscript can be with two minor comments that should be addressed.
1. The methods section should contain much greater detail, especially concerning
the pre-processing and analysis of EEG.
2. Information on IRB approval seems incomplete. This needs to be completed
and verified in a future review.

Author Response

Point 1. The methods section should contain much greater detail, especially concerning the pre-processing and analysis of EEG.

Response 1: We extended the EEG Methods section and devided into two subsections (EEG procedure and EEG analysis):

4.4. EEG procedure

The registration and evaluation of EEG was carried out in accordance with the IPEG guidelines. All recordings were obtained in the afternoon at 3-4 pm. EEGs were recorded for 3 min during resting, the subjects sitting comfortably in a chair. They were asked to close their eyes and to relax but stay awake during the recording. Prior to each recording, subjects were instructed to relax but stay awake. In order to keep constant the level of vigilance, an experimenter controlled on-line the subject and the EEG traces. He verbally alerted the subject any time there were signs of behavioral and/or EEG drowsiness.

EEG was recorded on Nihon Kohden 4217 G EEG using the time constant of 0.3 s. The high frequency cut-off was 45 Hz. The 14 Ag/AgCl electrodes were placed according to the international 10–20 system at O2, O1, P4, P3, C4, C3, F4, F3, Fp2, Fp1, T4, T3, F8, and F7 positions. Linked ears served as the reference. Electrode impedance did not exceed 10 kΩ. During the recording, 180 s of EEG at rest was simultaneously sampled at 256 Hz per channel and stored on a computer for further analysis offline. The EEG was reviewed visually for artifacts. Periods of artifact were eliminated from subsequent analysis. After the elimination of artifacts, segments of resting EEG of 120-s duration were selected for further analysis.

4.5. EEG analysis

Frequencies below 2 Hz and above 35 Hz were eliminated by digital filtering. Thirty 4-s epochs free of artifacts of resting EEG were processed by fast Fourier transform. The relative powers (% of total EEG power) of the delta (2.00–3.99 Hz), theta (4.00–7.99), alpha (8.00–12.99), beta1 (13.00–19.99), and beta2 (20.00–30.00) bands and for the regions: occipital (O2, O1), parietal (P4,P3), central (C4,C3), frontal 1 (F4, F3), frontal 2 (Fp2, Fp1), mid-temporal 2 (T4, T3) and anterior temporal (F8, F7) were calculated. Log transformations of the relative power of the various bandwidths in each derivation were calculated in order to compensate for data skewedness, as recommended by [51], using log [x/(1−x)], where x is the fraction of total power for each 4-s sample. The average log relative power for each frequency band was then calculated. The details of the spectral analysis procedures have been previously described [43].

 Point 2. Information on IRB approval seems incomplete. This needs to be completed and verified in a future review.

Response 2: The section IRB now contains all necessary information

Institutional Review Board Statement: The study was conducted according to the guidelines of the Declaration of Helsinki, and approved by The Research Center of Neurology Local Medical Ethics Committee ( â„– 11/14 19.11.2014) and Vavilov Institute of General Genetics Local Ethics Committee (â„– 4/24.11.2016).

Please see the attachment. All corrections are highlighted (yellow).
